# Prevalence and distribution of multilocus sequence types of *Staphylococcus aureus* isolated from bulk tank milk and cows with mastitis in Pennsylvania

**Asha Thomas**[¤a], **Shubhada Chothe, Maurice Byukusenge**[¤b]**, Tammy Mathews, Traci Pierre, Subhashinie Kariyawasam**[¤c]**, Erin Luley, Suresh Kuchipudi, Bhushan Jayarao** *

Penn State Animal Diagnostic Laboratory, Department of Veterinary and Biomedical Sciences, The Pennsylvania State University, University Park, Pennsylvania, United States of America

¤a Current address: Hematology/Oncology, School of Medicine, Case Western Reserve University, Cleveland, Ohio, United States of America
¤b Current address: School of Veterinary Medicine, University of Rwanda, Nyagatare, Rwanda
¤c Current address: Department of Comparative, Diagnostic, and Population Medicine, University of Florida, Gainesville, Florida, United States of America
* bmj3@psu.edu

**Data Availability Statement:** All relevant data are within the paper and its Supporting Information files.

## Abstract

A total of 163 *S. aureus* isolates; 113 from mastitic milk (MM) and 50 from bulk tank milk (BTM) (2008, 2013–2015) submitted for bacteriologic analysis at the Penn State Animal Diagnostic Laboratory were examined for their phenotypic and genotypic characteristics. Multi-locus sequence typing (MLST) analysis identified 16 unique sequence types (STs) which belonged to eight clonal complexes (CCs). Majority of the isolates were variants of CC97 (68.7%) and CC151 (25.1%). CC97 comprised of seven STs, of which two were new STs (ST3273, ST3274), while CC151 comprised of three STs of which ST3272 was identified for the first time. Several farms had more than one ST type that were either members of the same clonal complex or unrelated STs. On one farm, six different STs of both categories were seen over the years within the farm. It was observed that ST352 and ST151 were the two main clonal populations in cattle not only in Pennsylvania but also globally. Most isolates were susceptible to all the antibiotics evaluated. 6.7% of isolates showed resistance to vancomycin and penicillin. Two isolates of ST398 showed multidrug resistance (>3 antibiotics) against clindamycin, erythromycin, tetracycline, and penicillin. It was noted that 59 of 163 (36.2%) isolates encoded for enterotoxigenic genes. Enterotoxin genes *seg/sei* accounted for ~85% of enterotoxin positive isolates. Toxic shock syndrome gene *tsst-1* alone was positive in two isolates (ST352, ST 2187). 97.5% of CC151 isolates were enterotoxin *seg/sei* positive. Most isolates were positive for *lukED* (95%) and *lukAB* (96.3%) leukotoxin genes. Bovine specific bi-component leucocidin *lukMF'* was present in 54% of isolates. A prominent observation of this study was the explicit association of *lukMF'* with lineages ST151 and ST352. In conclusion, the findings of the study, suggest that small number of *S. aureus* STs types (ST352, ST2187, ST3028, and ST151) are associated with majority of cases of bovine mastitis in Pennsylvania dairy farms. It was observed that one ST of *S. aureus*

**Funding:** BJ Award #: 44166109 The Pennsylvania Department of Agriculture https://www.agriculture.pa.gov/Animals/AHDCommission/Pages/default.aspx The funders had no role in study design, data collection and analysis, decision to publish, or preparation of the manuscript.

**Competing interests:** The authors have declared that no competing interests exist.

predominated in the herd and this ST can coexist with several other ST types of *S. aureus* strains. When STs were interpreted along with virulence, leucocidin genes and antimicrobial resistance, ST-variants allowed better interpretation of the *S. aureus* molecular epidemiologic findings specifically for tracing recurrence or persistence of infections in cow over time, among cows in the herd, and between herds in Pennsylvania.

## Introduction

*Staphylococcus aureus* is a contagious mastitis pathogen that continues to be a challenge to a profitable dairy practice globally. This pathogen can cause various forms of mastitis, persist over time in mammary glands, elevate somatic cells, require the use of antimicrobials, cull chronically infected cows, lower milk production and quality of milk. Further, multidrug resistant and enterotoxigenic strains of *S. aureus* can cause serious illness in humans through consumption of contaminated or unpasteurized milk [1, 2].

Several intra- and inter- related interactions between the cow, *S. aureus* and the environment can dictate the severity and duration of *S. aureus* mastitis in lactating cows [3, 4]. Managing the cow and environmental related risk factors for *S. aureus* such as; 1) using blanket dry cow treatment, 2) pre- and post-milking teat disinfection, 3) monitoring teat condition, 4) wearing gloves during milking, 5) disinfect hands between handling of cows, 5) proper maintenance of milking machine, 6) cull infected cows, 7) ensuring stall hygiene and bedding; can reduce the frequency of *S. aureus* mastitis in lactating cattle [5, 6]. Herds that consistently employ proven mastitis control measures can reduce within herd *S. aureus* prevalence [5]. Sommerhäuser et al. [7] reported that traditional *S. aureus* control programs were found to be ineffective, as some *S. aureus* strains appeared to have similar characteristics to that of environmental mastitis pathogens.

Several recent studies have shown that *S. aureus* intra-mammary infections can vary between and within herds and this could be attributed to strain differences [8, 9]. *Staphylococcus aureus* strains encode and express a variety of virulence factors, which could explain for this pathogen's unique ability to cause varying types of infections in humans and animals including dairy cattle. Pathogenic strains of *S. aureus* possess a combination of toxins belonging to the three classes of toxins including pore-forming toxins, exfoliative toxins, and enterotoxins [10]. These toxins in concert with cell surface associated antigens and enzyme-based virulence factors could significantly add to the infectivity, virulence, and pathogenicity of *S. aureus* mastitis. *S. aureus* invade mammary epithelial cells, and in this intracellular environment seek shelter from exposure to antimicrobials and evade the host immune recognition and response system. Further in this intracellular environment *S. aureus* can persist and survive in the mammary gland for extended periods of time resulting in asymptomatic infection, and when host-pathogen balance is altered, *S. aureus* could remerge under favorable conditions and cause subclinical, clinical or chronic mastitis [11–13].

Multi-locus sequence typing (MLST), has been shown to be a highly reproducible typing method and has been widely used to determine the population structure, clonal and STs of *S. aureus* from milk and bovine mastitis [2]. Multi-locus sequence typing has allowed researchers to understand the population structure of *S. aureus* strains that occur not only in a given region of a country but also allow comparison to clonal types from around the world [14]. Hata et al. [15] reported that *S. aureus* CC97, CC126, CC133, CC479, and CC705 strains were associated with ruminants, while Schmidt, Kock and Ehlers [13] stated that *S. aureus* CC97 has

been reported from bovine mastitis cases in Africa, Asia, Europe and South America and in a few humans that had animal contacts.

Thus far, molecular characterization of bovine *S. aureus* isolates have shown that; 1) a small number of *S. aureus* clonal types are associated with majority cases of bovine mastitis globally [16, 17], 2) *S. aureus* strains can be herd-specific [9], 3) *S. aureus* strains can be associated with varied clinical consequences in the lactating cow with respect to severity and persistence of infections [18, 19], and 4) in most herds, one pathogenic strain of *S. aureus* predominates and this strain can coexist with several other *S. aureus* strains [20, 21].

The purpose of the present study was to screen 163 isolates of *S. aureus* from 77 herds in Pennsylvania that were isolated from 2008–2015 from quarter milk and BTM samples sent for bacteriologic testing to the Penn State Animal Diagnostic Laboratory. The *S. aureus* isolates were examined for; 1) resistance to antimicrobials, 2) virulence genes, and 3) distribution of MLST sequence and clonal types. The findings of the study will allow identification of strains with unique phenotypic and genotypic characteristics associated with persistence in a cow, herd and among herds in Pennsylvania, which would later be used to develop tailored *S. aureus* mastitis prevention and control practices to address *S. aureus* mastitis.

## Materials and methods

### Bacterial isolates

The 163 isolates examined in this study were from 2008, and 2013–2015 isolated from cows with mastitis (n = 113 isolates) and from BTM (n = 50) from 77 dairy herds in Pennsylvania.

This study was carried out in accordance with the requirements of the Pennsylvania State University animal care and use committee and did not require IACUC approval. The milk samples were collected and submitted to the Penn State Animal Diagnostic Laboratory by licensed practicing veterinarians in Commonwealth of Pennsylvania for routine diagnostic mastitis milk culture.

*Staphylococcus aureus* isolates from these milk samples were placed in a culture repository. Freezer stocks of previously identified isolates were sub-cultured from -80˚C culture stocks onto Tryptic Soy Agar (TSA) plate with 5% sheep blood (Remel, Inc., Lexena, KS) and incubated aerobically for 24–48 h at 37˚C. The isolates were examined for hemolysis on blood agar, Gram reaction, morphology, and catalase reaction and speciated using MALDI-TOF MS species identification system (Bruker Daltonics, Bremen, Germany) as described previously [22]. The isolates were subcultured each week onto TSA plate with 5% sheep blood (Remel) for further characterization.

### Antimicrobial susceptibility testing

The agar disk diffusion assay was performed and interpreted as described by CLSI [23]. Ten antimicrobials including 10 units of Penicillin (P), 30 μg of Amoxicillin-Clavulanic acid (AMC), 30 μg Cefoxitin (FOX), 5 μg of Ciprofloxacin (CIP), 2 μg of Clindamycin (DA), 15 μg of Erythromycin (E), 10 μg of Gentamicin (CN), 1 μg of Oxacillin+ 2% NaCL (OX), 30 μg of Tetracycline (TE) and 30 μg of Vancomycin (VA) diffusion disks (Oxoid, Thermofisher Scientific Inc., Waltham, MA) were used to determine antimicrobial susceptibility/resistance of *S. aureus*. *Staphylococcus aureus* ATCC 25923 strain was used as the control strain.

### Genotypic characterization

**DNA extraction.** One to two colonies of *S. aureus* from a TSA plate with 5% sheep blood (Remel) were transferred to a Brain Heart Infusion Broth (BD Diagnostics, Sparks, MD) and

incubated overnight (16 h) at 37˚C. One ml of overnight culture was used for DNA extraction. Bacterial cells were pelleted by centrifugation and washed once with 1 ml of TE buffer (10 mM Tris and 5 mM EDTA; pH 7.8). The bacterial pellet was suspended in 200 µl of InstaGene purification matrix (InstaGene Matrix, Bio-Rad, Hercules, CA). The mixture was incubated at 56˚C for 30 min followed by high speed vortexing for 10s and 10-min incubation at 100˚C. The cell lysate was vortexed and then centrifuged for 5 min at 14,000 rpm. The supernatant was collected, and DNA was quantified using a NanoDrop Lite spectrophotometer (Nano-Drop Technologies, CA) and stored at −20˚C until use. The DNA samples diluted to 100 ng/µl were used for PCR assays.

**Detection of genes that encode for virulence factors.** Isolates of *S. aureus* were examined for genes that encode for enterotoxins (*sea*, *seb*, *sec*, *sed*, *see*, *seg*, *seh*, *sei*, *sej*, *sek*, *sel*, *sem*, *sen*, *seo*, *sep*, *seq*, *ser*), and the toxic shock syndrome toxin (*tsst-1*) using primer sets and conditions described previously [24]. The primers set and conditions for enterotoxins and *tsst-1* were optimized for a multiplex PCR assay kit (QIAGEN multiplex PCR kit, QIAGEN), while genes for leukocidins, *lukAB*, *luk MF and lukED* were assayed using standard PCR reagents (Thermofisher Scientific Inc., Waltham, MA) as described previously [25–27]. The primers set and positive control strains used for PCR assays are shown in S1 Table.

**Multilocus sequence typing.** Multi locus sequence typing was done as described in the *S. aureus* MLST database (https://pubmlst.org/organisms/staphylococcus-aureus) and elsewhere [28]. Briefly primers provided at the *S. aureus* MLST database were used to amplify seven housekeeping genes (arcC, aroE, glpF, gmk, pta, tpi and yqiL) (S1 Table). The PCR products were purified using ChargeSwitch® PCR Clean-Up Kit (Invitrogen, CS12000) and sequenced at Penn State Genomics Core Facility. The sequence chromatograms were analyzed for quality and trimmed manually. The allelic number of the genes and ST of each isolate were assigned by uploading the sequences at the *S aureus* MLST website (https://pubmlst.org/organisms/staphylococcus-aureus/submissions). Novel allele sequences and STs were determined by sending the trace files of respective genes/sample to database curator.

The minimum spanning trees for *S. aureus* STs from Pennsylvania and The United States and distribution of *lukMF'* gene among STs from Pennsylvania were generated using the MSTree V2 algorithm implemented in the GrapeTree program (https://github.com/achtman-lab/GrapeTree). Probable pattern of evolutionary decent based on allelic profile was also visualized using Grapetree program (https://github.com/achtman-lab/GrapeTree). The occurrence of *S. aureus* STs from Pennsylvania milk samples were compared to STs from milk samples from other parts of the world using Global optimal eBURST (goeBURST, http://www.phyloviz.net/goeburst/) algorithm in PHYLOViZ software [29].

The PHYLOViZ software [29] was also used to obtain allelic frequency and Simpson's Index of Diversity. Nucleotide diversity, number of segregating sites and Tajima's D were analyzed with MEGA v7.0.25 [30]. Non-synonymous to synonymous substitution (dN/dS) mutation at the MLST loci was calculated via Nei and Gojobori method using START2 software [31]. The dN/dS ratio < 1 indicate purifying dN/dS, dN/dS = 1 shows neutral selection and dN/dS > 1 indicate positive selection during evolution. The extent of clonality and recombination in the population was assessed by calculating the Standardized Index of Association ($I^S_A$) executed in START2 [31].

**Phylogenetic analysis.** Phylogenetic analysis was done by analyzing concatenated in-frame sequences for each unique ST (3186 bp) which was downloaded from *S. aureus* MLST site (https://pubmlst.org/organisms/staphylococcus-aureus). The sequences were translated to protein sequences using ExPASy (http://web.expasy.org/translate/). Both concatenated nucleotide and amino acid sequences were aligned in ClustalX as well as using Muscle in MEGA v7.0.25 and was used for phylogenetic analysis using MEGA v7.0.25. Phylogenetic trees

utilizing both the sequences were constructed using maximum likelihood (ML) (Tamura-Nei and Poisson model for amino acid) and UPGMA (not shown) (Kimura's 2- parameter model for nucleotide and Poisson model for amino acid) with boo*t st*rap analysis (500 replications) using MEGA v7.0.25.

## Results

### Distribution of clonal and sequence types associated with MM and BTM samples

The 163 isolates belonged to eight CCs and 16 STs, of which CC97, and CC151 accounted for 68.7% and 24.5% of the isolates, respectively. *Staphylococcus aureus* ST1, was isolated from a BTM, while ST15, ST45, and ST72 were isolated from cows with mastitis. The *S. aureus* isolates that belonged to CC97 comprised of seven STs, three of the seven STs (352, 2187, 3028) were isolated from both cows with mastitis and BTM, while ST97 and ST693 were isolated from cows with mastitis. Two unique STs, ST3273 and ST3274 of *S. aureus* were isolated from BTM, both of which to date have not been reported in literature. The most frequent ST type was ST352 which accounted for 27% (44 of 163) of all the isolates followed by ST2187 (n = 28, 17.2%) and ST3028 (n = 28, 17.2%). Three STs (ST151, ST705, and ST3272) belonged to CC151 (n = 40, 24.5%). ST151 (n = 37, 23%) was isolated from cows with mastitis and BTM, while *S. aureus* ST705, and ST3272 (which we identified for the first time) was isolated from cows with mastitis. Sequence type 350, and ST398 were isolated from MM and BTM, respectively (Fig 1, Table 1).

The GrapeTree algorithm was used to determine population distribution of *S. aureus* isolates from Pennsylvania dairy farms. The GrapeTree analysis revealed that the majority of the STs were grouped into two CCs, CC97 and CC151 (Fig 1). The second clonal complex was CC151 (n = 41, 25.1%) with two single locus variants of which ST3272, we report for the first time (Fig 1).

Comparison of our data to the *S. aureus* database for isolates from bovine milk in USA (https://pubmlst.org/organisms/staphylococcus-aureus) showed that ST352 and ST151 are the two major clonal populations reported in USA (Fig 2). We also compared our data to the

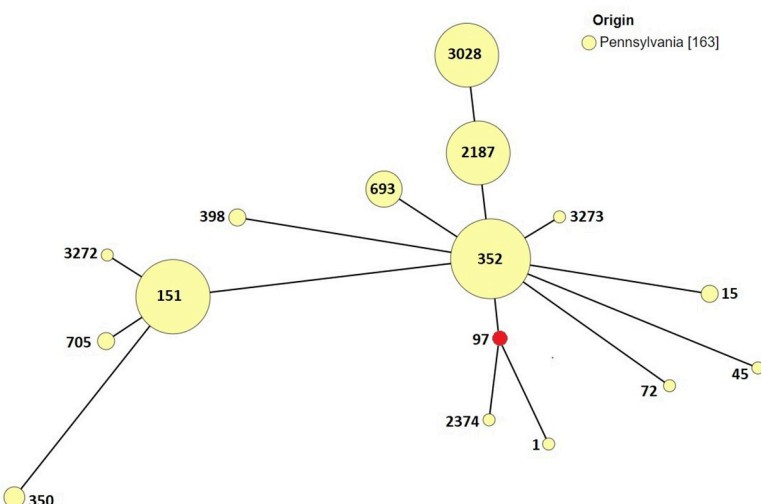

**Fig 1. Minimum spanning tree of *S. aureus* isolates from Pennsylvania dairy farms.** Each node represents a ST. CC97 is the primary founder (shown in red). The size of the nodes denotes the sample size.

**Table 1. Distribution of clonal complexes, sequence types, antimicrobial sensitivity, staphylococcal toxins, and leucocidin profiles of *S. aureus* isolated from mastitic and bulk tank milk samples from dairy herds in Pennsylvania in 2008, 2013–2015.**

| Clonal Complex | Sequence Type | Mastitic Milk (n = 113) | Bulk Tank Milk (n = 50) | % Isolates | Antimicrobial Sensitivity [a, dd] | Staphylococcal Toxin Gene Profiles [b, d] | Leucocidin Profiles [c, d] |
|---|---|---|---|---|---|---|---|
| | | No. (%) Isolates | | | | | |
| 1 | 1 | - | 1 (2) | 0.61 | - | *seh* (1) | *lukAB-lukED* (1) |
| 15 | 15 | 2 (1.8) | - | 1.22 | PEN | - | *lukAB-lukED* (2) |
| 45 | 45 | 1 (0.9) | - | 0.61 | PEN | *seg-sei-sem-sen-seo* (1) | *lukAB* (1) |
| 72 | 72 | 1 (0.9) | - | 0.61 | PEN | *seg-sei-sem- sen-seo-seq* (1) | *lukAB-lukED* (1) |
| 97 | 97 | 1 (0.9) | - | 68.71 | PEN | *sec-sed-sej-tsst-1-sel-ser* (1) | *lukMF'-lukAB-lukED* (1) |
| | 352 | 25 (22.1) | 19(38) | | - (35), TET (5), FOX (1), PEN (2), VA (1) | -(35); *sed* (1), *seg* (1), *seq* (1), *tsst-1* (1), *seg-sei* (4); *sec-sed-sej-tsst-1-sel-ser* (1) | - (1), *lukAB-lukED* (1), *lukMF'-lukED* (2), *lukMF'-lukAB-lukED* (40) |
| | 693 | 9 (8.0) | - | | - | -(9) | *lukAB-lukED* (9) |
| | 2187 | 20 (17.8) | 7 (14) | | - (24), FOX (1), VA (3) | -(26); *tsst-1* (1); *seg-sei* (1) | - (1), *lukAB* (1), *lukED* (1), *lukAB-lukED* (25) |
| | 3028 | 16 (14.1) | 14 (28) | | - (25), VA (3) | -(27); *seg-sei* (1) | *lukAB-lukED* (25) *lukMF'-lukAB-lukED* (3) |
| | 3273 | - | 1 (2) | | - | - | *lukMF'-lukAB-lukED* (1) |
| | 3274 | - | 1 (2) | | PEN | - | *lukAB-lukED* (1) |
| 151 | 151 | 31 (27.4) | 6 (12) | 24.53 | - (31), VA (3), DA (1), DA, ERY (2) | -(1); *seg-sei* (34); *sed-seg-sei* (1); *seg-sen-sei* (1) | *lukAB-lukED* (1) *lukMF'-lukAB-lukED* (36) |
| | 705 | 2 (1.8) | - | | - (1), PEN (1) | *sec-seg-sei-tsst-1-sel* (2) | *lukMF'-lukAB-lukED* (2) |
| | 3272 | 1 (0.9) | - | | - (1) | *seg-sei* (1) | *lukMF'-lukAB-lukED* (1) |
| 350 | 350 | 3 (2.7) | - | 1.84 | - (3) | *sei-sem* (2), *seg-sei-sem* (1) | *lukAB* (2), *lukMF'-lukAB-lukED* (1) |
| 398 | 398 | 1 (0.9) | 1 (2) | 1.22 | DA, ERY, TET, PEN (2) | -(2) | - (1), *lukAB* (1) |

[a] **Antimicrobial resistance**: Pan-susceptible (80.3%), Cefoxitin (FOX, 1.84%), Clindamycin (DA, 2.4%), Erythromycin (ERY, Penicillin (PEN, 6.74%), Tetracycline (TET, 4.9%), Vancomycin (VA, 6.13%).

[b] **Staphylococcal toxin genes**: Toxin genes detected in 64.6% (*sec*, 2%; *sed*,2%; *seg*,29%; *seh*,1%; *sei*,29%; *sej*,1%; *sel*,2%; *sem*,3%; *sen*, 1%; *seo*, 1%; *seq*, 1%; *ser*, 1%; *tsst-1*,3%) isolates.

[c] **Leucocidin genes**: Leucocidin positive isolates detected in 98.15% (*lukMF'*, 54%; *lukAB2*, 96.3% and *lukED*, 95%); Profiles: *LukAB2* (3%), *lukED* (0.6), *lukAB2-lukED* (40.5%), *lukMF'- lukED* (1.2%), *lukMF'-lukAB2-lukED* (52.7%) isolates.

[d] No. of isolates.

*S. aureus* STs in the world database for isolates from bovine milk (Fig 3). As observed with our study, CC97 is also the major clonal population of *S. aureus* reported from bovine milk samples worldwide. CC151 is a rapidly evolving clonal complex of which ST705 is a subgroup founder. CC1 from our data also represents an evolving bovine associated *S. aureus* group in the world (Figs 2 and 3).

## Phylogenetic analysis

Although analysis focused on nucleotide changes illustrate the evolutionary pathway, amino acid sequences are more likely to provide more information about the adaptation of a new clonal population since it eliminates all neutral (non-synonymous) substitution mutations. In our study, we identified a new ST, ST3273 based on the stop codon at *glp* locus (allele 431, 11th codon). This suggests that the mutation that was observed is not deleterious or there is redundancy in the glycerol kinase enzyme gene that is encoded by the gene. Phylogenetic trees constructed by Maximum likelihood method using concatenated nucleotide and amino acid sequences (Fig 4), both showed clustering of clonal complexes similar to GrapeTree analysis.

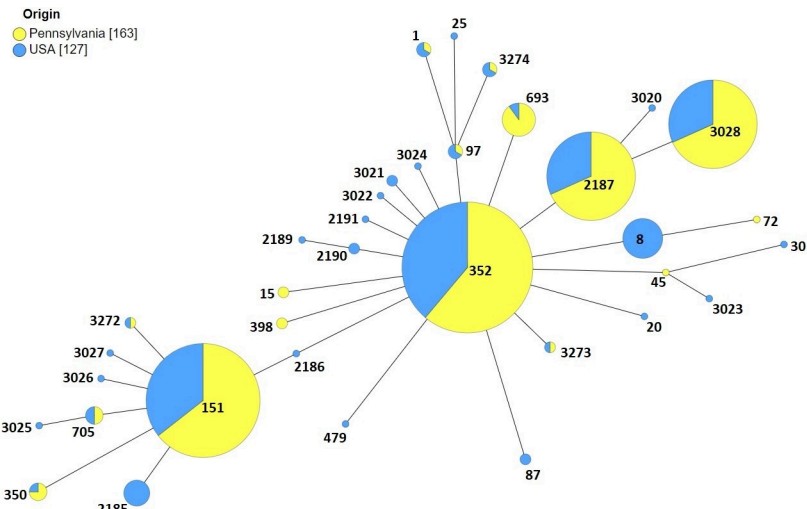

**Fig 2. Prevalence of *S. aureus* STs isolated from milk samples in Pennsylvania and the United States shown as minimum spanning tree.** Each node represents a ST. For each ST, the yellow color represents the proportion of samples from Pennsylvania (this study). Nodes with the blue color only denotes STs that were previously identified in the USA milk samples but were not found in our study while nodes with the yellow color only represent STs that were found in Pennsylvania milk samples but were not previously reported in the US. The size of the nodes denotes the sample size.

SLVs of the two clonal groups (CC151 and CC97) identified in this study were tightly clustered while DLV and other variants were located distantly from the main cluster in a sequential manner (Fig 2).

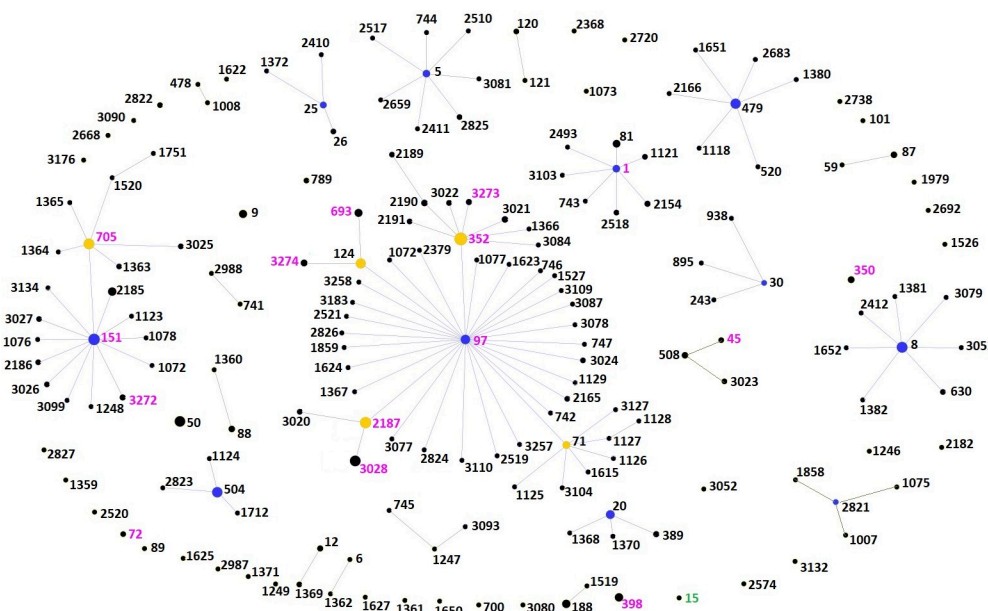

**Fig 3. Comparison of the occurrence of *S. aureus* STs isolated from Pennsylvania milk samples to STs from milk samples from other parts of the world using goeBURST analysis.** Each node represents an ST. The size of the node denotes sample size. The primary founder is shown in blue and subgroup founder is shown in yellow. STs found only in the world milk sample data are in labelled black. STs found in both world milk data and out dataset are labeled pink and ST15 found only in our dataset is labeled green.

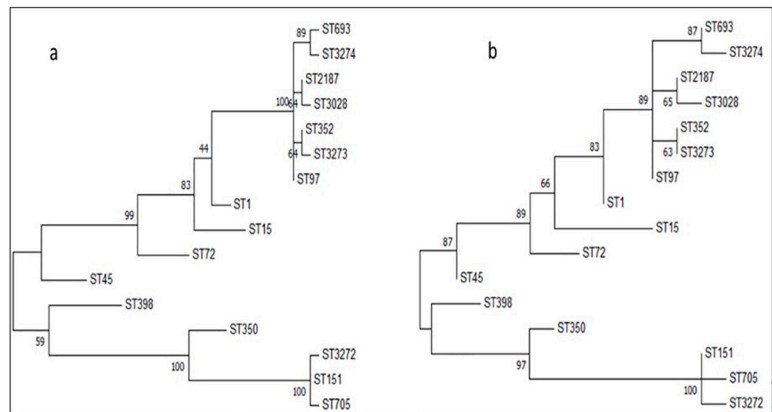

**Fig 4. Phylogenetic analysis of *S. aureus* sequence types using maximum likelihood method.** Phylogenetic trees were derived using concatenated (a) nucleotide and (b) amino acid sequences of seven housekeeping genes of *S. aureus*. Isolates of *S. aureus* cluster around two branches; the clustering of isolates on nucleotide sequence analysis is similar to that observed on amino acid sequence analysis. Bootstrap values (500 replications) for both trees are shown at the divergence points of each of the respective branches on the tree.

## Sequence type diversity of *S. aureus*

Number of alleles observed for each locus varied from five (arcc) to ten (aroe). Simpson's diversity index which measures the ability of a locus to discriminate related/unrelated isolates showed high discriminatory power for aroe (0.93) while the lowest was for glp (0.69). Number of segregating sites (S- any site that shows 2 or more nucleotides among *n* sequences) was highest for aroe (23) and lowest for arcc (6) based on the nucleotide sequence. Amino acid sequences had highest number of segregating sites in aroe locus (11) itself (indicating non-synonymous substitution mutations) and lowest in glp (2). For the yqil locus, 19 segregating sites in nucleotide sequence translated to only 5 in amino acid sequence (synonymous substitutions). Nucleotide diversity (π- average number of nucleotide differences per site) also showed similar result. Tajima's D values which compare the number of segregating sites per site with the nucleotide diversity were not statistically significant for any of the genes indicating evolution by random process. Non-synonymous to synonymous mutation ratios for all the loci was <1 true of purifying selection during evolution (Table 2).

## Clonality of *S. aureus*

Standardized Index of Association ($I^S_A$) was used to calculate the degree of linkage between the gene alleles in a population. A highly clonal population will have an $I^S_A$ value of 1 indicating strong linkage between the loci/alleles. A value of zero is associated with a freely recombining, non-clonal population. The $I^S_A$ for the *S. aureus* examined in this study (163 isolates comprising of 16 STs) was determined to be 0.7339 and 0.4201 indicating significant (p = 0.000) linkage disequilibrium, substantiating the clonal nature of *Staphylococcus aureus*.

## Antimicrobial resistance

Majority of *S. aureus* isolates (n = 130; 80.3%) were susceptible to all antibiotics evaluated. Antimicrobial resistance was observed to cefoxitin (n = 3, 1.84%), clindamycin (n = 4, 2.45%), erythromycin (n = 2, 1.84%) penicillin (n = 11, 6.74%), tetracycline (n = 6, 3.68%) and vancomycin (n = 11, 6.74%). Two isolates of ST398 showed multidrug resistance (>3 antibiotics) against clindamycin, erythromycin, tetracycline, and penicillin (Table 1). Five (ST15, ST45,

**Table 2. Nucleotide and allelic diversity of MLST loci of *S. aureus* clones isolated from milk samples in Pennsylvania dairy herds.**

| Locus | Allele No. | Simpson's Diversity Index | No. Segregating sites[a] | | Nucleotide Diversity[b] | | Tajima's Test Statistic[c] | | Non-synonymous to synonymous mutation ratio[d] |
|---|---|---|---|---|---|---|---|---|---|
| | | | $S^n$ | $S^p$ | $\pi^n$ | $\pi^p$ | $D^n$ | $D^p$ | dN/dS |
| ARCC | 5 | 0.708 | 6 | 3 | 0.007 | 0.009 | 0.286 | 0.175 | 0.260 |
| AROE | 10 | 0.933 | 23 | 11 | 0.015 | 0.018 | -0.814 | -1.380 | 0.180 |
| GLP | 7 | 0.692 | 8 | 2 | 0.006 | 0.005 | -0.809 | -0.275 | 0.198 |
| GMK | 8 | 0.842 | 13 | 5 | 0.011 | 0.011 | -0.302 | -0.923 | 0.129 |
| PTA | 9 | 0.817 | 10 | 5 | 0.008 | 0.011 | 0.124 | -0.142 | 0.270 |
| TPI | 8 | 0.800 | 14 | 4 | 0.013 | 0.010 | -0.039 | -0.322 | 0.112 |
| YQIL | 7 | 0.742 | 19 | 5 | 0.015 | 0.013 | 0.108 | 0.594 | 0.150 |
| Concatenated Sequence | 16 | 1 | 93 | 35 | 0.009 | 0.010 | 0.061 | 0.008 | - |

[a] S = Number of segregating sites, [n]- Nucleotide sequence, [p]- amino acid sequence.

[b] $\pi$ = Nucleotide diversity, [n]- Nucleotide sequence, [p]- amino acid sequence.

[c] D = Tajima test statistic, [n]- Nucleotide sequence, [p]- amino acid sequence.

[d] dN/dS = Non-synonymous to synonymous mutation ratio.

ST72, ST97, ST705, ST3274) of the 16 STs were resistant to penicillin and these isolates were from cows with mastitis. No unique distribution of antimicrobial resistance profiles was observed between mastitis and BTM isolates (Tables 1, 3–5).

## Toxin type gene profiles

A total of 163 isolates were examined for enterotoxins of which 105 (64.41%) isolates encoded for potential enterotoxigenic genes. Genes of enterotoxin *egc*-cluster (*seg*, *sei*, *sem*, *seo*, and *sen)* were the most frequently identified enterotoxin genes (Table 1). The presence of SaPIbov pathogenicity island that encodes for *sec*/*tsst-1*/*sel* enterotoxins was identified in four isolates (ST97, ST352 and ST705). Toxic shock syndrome gene *tsst-1* was present in five isolates (ST97, ST352, ST705, ST2187). Classic enterotoxin genes, including *sea* and *seb* were not detected in any of the samples while *sed* was detected in three isolates. Enterotoxins *seh*, *seq*, *sej*, *ser* and *seg* alone were present in one isolate each. 39 of 40 (97.5%) of CC151 isolates encoded for *seg*/*sei*, while isolates in the CC97 were mostly negative for enterotoxins. Interestingly, the *egc*-cluster genes *seg*/*sei* were confined to CC151 (Tables 4 and 5). The entire *egc*-cluster was detected in ST45 and ST72 isolates (Table 3). Eight of 50 (16%) BTM isolates encoded for toxin genes, while 46 of 58 (79.3%) isolates from cows with mastitis encoded for two or more toxin genes (Tables 1, 3, 4 and 5).

## Leucocidin gene profiles

A total of 163 isolates were examined for leucocidins *lukMF'*, *lukAB*, and *lukED* of which 160 of 163 (98.15%) encoded for one or more leucocidins. Leucocidins *lukMF'*, *lukAB*, and *lukED* were present in 54, 96.3 and 95% of the isolates, respectively (Table 1). Bovine specific bi-component leucocidin *lukMF'* was associated with ST97, ST352, ST3273, ST151, ST3272 and ST705 (Fig 5). *LukMF'* was detected either in combination with *lukAB* or *lukED* or both (Table 1). LukAB was detected in five isolates belonging to four STs, while a single isolate encoded for *lukED* belonging to ST2187. *LukAB* and *LukED* was detected in 66 of 163 (40.5%) isolates belonging to nine STs, while two isolates encoded for *lukMF'* and *lukED* belonging to

**Table 3. Distribution and prevalence of variants *Staphylococcus aureus* sequence types of clonal complex 97 of isolated from 2008, and 2013–2015 from dairy herds in Pennsylvania.**

| Sequence Type profile[a] | Year | No. of herds | Sample [b] | Anti-microbial resistance[c] | Toxin gene profile | Leucocidin profile |
|---|---|---|---|---|---|---|
| 97 | 2008 | 1 | MM (1) | PEN | *sec, sed, sej, tsst-1, sel, ser* | *lukMF'-lukAB-lukED* |
| 352–1 | 2008 | 5 | MM (5), BTM (1) | - | - | *lukMF'-lukAB-lukED* |
| | 2013 | 4 | MM (5), BTM (1) | | | |
| | 2014 | 3 | MM (1), BTM (2) | | | |
| | 2015 | 11 | MM (4), BTM (7) | | | |
| 352–2 | 2008, 2015 | 2 | MM (2) | - | *seg, sei* | *lukMF'-lukAB-lukED* |
| 352–3 | 2008 | 1 | MM (1) | FOX | - | *lukMF'-lukAB-lukED* |
| 352–4 | 2008 | 1 | MM (1) | - | *tsst-1* | *lukMF'-lukAB-lukED* |
| 352–5 | 2008 | 1 | MM (1) | - | *sec, tsst-1, sel* | *lukMF'-lukAB-lukED* |
| 352–6 | 2008 | 1 | MM (1) | TET | *seq* | *lukMF'-lukAB-lukED* |
| 352–7 | 2014, 2015 | 3 | MM (3), BTM (1) | TET | - | *lukMF'-lukAB-lukED* |
| 352–8 | 2014 | 2 | BTM | VA | - | *lukMF'-lukAB-lukED* |
| 352–9 | 2014, 2015 | 2 | BTM (2) | - | - | *lukM-lukED* |
| 352–10 | 2015 | 1 | BTM | PEN | - | *lukMF'-lukAB-lukED* |
| 352–11 | 2015 | 1 | BTM | PEN | *seg, sei* | *lukMF'-lukAB-lukED* |
| 352–12 | 2015 | 1 | BTM | - | *sed* | *lukMF'-lukAB-lukED* |
| 352–13 | 2015 | 1 | BTM | - | - | *lukAB-lukED* |
| 352–14 | 2015 | 1 | MM (1) | - | - | - |
| 2187–1 | 2008 | 2 | MM (2) | - | - | *LukAB-LukED* |
| | 2013 | 3 | MM (10) | | | |
| | 2014 | 2 | BTM (4) | | | |
| | 2015 | 4 | MM (3), BTM (1) | | | |
| 2187–2 | 2008 | 1 | MM (1) | VA | - | - |
| 2187–3 | 2008 | 1 | MM (1) | - | *tsst-1* | *LukAB-lukED* |
| 2187–4 | 2008 | 1 | MM (1) | - | *seg, sei* | *LukAB-lukED* |
| 2187–5 | 2013 | 1 | MM (1) | VA | - | *LukAB-lukED* |
| 2187–6 | 2014 | 1 | MM (1) | - | - | *LukED* |
| 2187–7 | 2014 | 1 | BTM | FOX | - | *LukAB-lukED* |
| 2187–8 | 2014 | 1 | BTM | VA | - | *LukAB* |
| 3028–1 | 2008, 2015 | 3 | MM (3) | - | - | *LukMF'-LukAB-lukED* |
| 3028–2 | 2008 | 76 | MM (1) | - | *seg, sei* | *LukAB-lukED* |
| 3028–3 | 2008 | 11 | MM (1) | - | - | *LukAB-lukED* |
| | 2013 | 4 | MM (5) | | | |
| | 2014 | 5 | MM (5), BTM(8) | | | |
| | 2015 | 6 | MM (1) BTM (3) | | | |
| 3028–4 | 2014 | 2 | BTM (3) | VA | - | *LukAB-lukED* |
| 693 | 2014 | 1 | MM (9) | | | *LukAB-lukED* |
| 3273 | 2014 | 1 | BTM | | | *LukMF'-LukAB-lukED* |
| 3274 | 2014 | 1 | BTM | PEN | | *LukAB-lukED* |

[a] Sequence Type Profile: composite profile including ST, antimicrobial resistance, toxin genes and Leucocidin genes (e.g., ST351-11; PEN, *seg, sei, lukMF'-lukAB-lukED*).

[b] Sample: MM; Mastitic milk, BTM; Bulk Tank Milk.

[c] Antimicrobial: FOX, Cefoxitin; DA Clindamycin; ERY, Erythromycin; PEN, Penicillin; TET, Tetracycline; VA, Vancomycin.

**Table 4. Distribution and prevalence of variants of *Staphylococcus aureus* sequence types of clonal complex 151 isolated from 2008, and 2013–2015 from dairy herds in Pennsylvania.**

| Sequence Type profile[a] | Year | No. of herds | Sample [b] | Anti-microbial resistance[c] | Toxin gene profile | Leucocidin profile |
|---|---|---|---|---|---|---|
| 151–1 | 2008 | 3 | MM (3) | - | *seg, sei* | *LukMF'-LukAB-lukED* |
| | 2013 | 2 | MM (3) | | | |
| | 2014 | 1 | BTM (2) | | | |
| | 2015 | 6 | MM (16), BTM (4) | | | |
| 151–2 | 2008 | 2 | MM (2) | VA | *seg, sei* | *LukMF'-LukAB-lukED* |
| 151–3 | 2008 | 1 | MM (1) | DA | *seg, sei* | *LukMF'-LukAB-lukED* |
| 151–4 | 2008 | 1 | MM (1) | - | *seg, sei* | *LukAB-lukED* |
| 151–5 | 2008 | 1 | MM (1) | - | *seg, sei, sen* | *LukMF'-LukAB-lukED* |
| 151–6 | 2015 | 1 | MM (1) | - | *sed, seg, sei* | *LukMF'-LukAB-lukED* |
| 151–7 | 2015 | 1 | MM (2) | DA, ERY | *seg, sei* | *LukMF'-LukAB-lukED* |
| 151–8 | 2015 | 1 | MM (1) | VA | - | *LukMF'-LukAB-lukED* |
| 705–1 | 2008 | 1 | MM (1) | - | *sec, seg, sei, tsst-1, sel* | *LukMF'-LukAB-lukED* |
| 705–2 | 2014 | 1 | MM (1) | PEN | *sec, seg, sei, tsst-1, sel* | *LukMF'-LukAB-lukED* |
| 3272 | 2008 | 1 | MM (1) | - | *seg, sei* | *LukMF'-LukAB-lukED* |

[a] Sequence Type profile: composite profile including ST, antimicrobial resistance, toxin genes and leucocidin genes (e.g., ST151-2; VA, *seg, sei, lukMF'-lukAB-lukED)*.

[b] Sample: MM; Mastitic milk, BTM; Bulk Tank Milk.

[c] Antimicrobial: DA Clindamycin; ERY, Erythromycin; PEN, Penicillin; VA, Vancomycin.

ST352. All three leucocidins (*lukMF'*, *lukAB*, and *lukED*) was detected in 86 of 163 (52.7%) of isolates belonging to eight STs (Tables 1, 3–5).

## Prevalence and distribution of sequence types of *S. aureus* in dairy herds in Pennsylvania

Sequence types ST151, ST352, ST2187 and ST3028 were the most prevalent STs and all of the four STs were from *S. aureus* isolates from 2008, 2013, 2014 and 2015 from MM and BTM samples from dairy herds in Pennsylvania (Tables 3 and 4). Combining STs, with toxin and leucocidin gene profiles and antimicrobial resistance profiles resulted in identifying ST-variants.

A total of 44 isolates from 22 dairy herds were identified as ST352, of which 25 and 19 isolates were from MM milk and BTM, respectively. Sequence type 352 was categorized into 14

**Table 5. Distribution and prevalence of Infrequent *S. aureus* clonal complexes Isolated from 2008, and 2013–2015 from dairy herds in Pennsylvania.**

| Clonal Complex (Sequence Type) | Year | No. of herds | Sample[a] | Anti-microbial resistance[b] | Toxin gene profile | Leucocidin profile |
|---|---|---|---|---|---|---|
| CC1 (ST1) | 2015 | 1 | BTM | - | *seh* | *lukAB-lukED* |
| CC15 (ST15) | 2013 | 1 | MM (2) | PEN | - | *lukAB-lukED* |
| CC45 (ST45) | 2008 | 1 | MM (1) | PEN | *seg, sei, sem, seo, sen,* | *lukAB* |
| CC72 (ST72) | 2015 | 1 | MM (1) | PEN | *seg, sei, sem, seo, sen, seq* | *lukAB-lukED* |
| CC350 (ST350) | 2008 | 1 | MM (1) | - | *sei, sem* | *lukAB* |
| | 2014 | 1 | MM (1) | - | *sei, sem* | *lukAB* |
| | 2015 | 1 | MM (1) | - | *seg, sei, sem* | *lukMF'-lukAB-lukED* |
| CC398 (ST398) | 2014 | 1 | MM (1) | DA, ERY, TET, PEN | - | - |
| | 2015 | 1 | BTM | DA, ERY, TET, PEN | - | *lukAB* |

[a] Sample: BTM, bulk tank milk; MM, Mastitic milk.

[b] Antimicrobial: DA Clindamycin; ERY, Erythromycin; PEN, Penicillin; TET, Tetracycline.

ST-variants based on antimicrobial resistance, toxin gene and leucocidin profiles. Subtype ST352-1 was the most predominant variant (35 isolates, 79.5%) and was present in *S. aureus* isolates from 2008 (5 herds), 2013 (4 herds), 2014 (2 herds), and 2015 (11 herds). Isolates identified as ST352-1 were susceptible to all antimicrobials tested, did not encode for any of the toxin genes but carried all three leucocidins genes (*lukMF', lukAB and LukED*). The remainder of the 18 isolates belonged to ST352-2 through ST352-14 variants. ST352-2 variant was observed on two herds one in 2008 and 2015, both isolates were from cows with mastitis, this variant was similar to that of ST352-1 with the exception that both of these isolates encoded for *seg* and *sei* enterotoxin genes. Variant ST325-3 was similar to ST352-1 but was also resistant to cefoxitin. Variants ST352-4, ST352-5 harbored *tsst-1* gene, in addition ST352-5 also encoded for *sec* and *sel* enterotoxin genes. ST352-6 was isolated from a cow with mastitis, this was the only isolate that harbored *seq* gene and was also resistant to tetracycline. ST352-7 was identified in 2014 (2 herds) and 2015 (1 herd), this subtype was similar to ST352-1 with the exception that all of the four isolates were resistant to tetracycline (Table 3).

ST352-9 did not encode for toxin genes and *lukAB* but was found in BTM in 2014 and 2015 from two different herds respectively. Variants ST352-10 and ST352-11 were similar except that ST352-11 encoded for *seg* and *sei* genes. ST352-12 was the isolate that was resistant to vancomycin. ST352-13 was isolated from BTM and encoded for enterotoxin *sed*. Interestingly subtype ST352-14 was susceptible to all antimicrobials and did not encode for toxin nor leucocidin genes (Table 3).

A total of 28 isolates from 14 dairy herds were identified as ST2187, of which 21 and seven isolates were from MM and BTM, respectively. It was observed that none of the 28 isolates encoded for *lukMF'* gene (Fig 5). ST2187 were categorized into 8 ST-variants based on antimicrobial resistance, toxin gene and leucocidin profiles. Variant ST2187-1 was the most predominant subtype (21 isolates, 75%) and was present in *S. aureus* isolates from 2008 (2 herds), 2013 (3 herds), 2014 (2 herds), and 2015 (4 herds). Isolates identified as ST2187-1 were susceptible

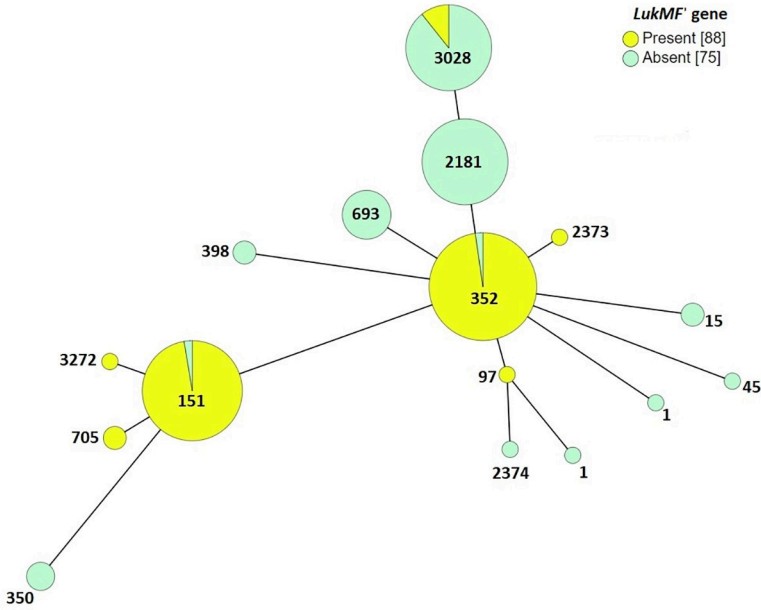

**Fig 5. Minimum spanning tree depicting distribution of leukocidin *lukMF'* gene among STs from Pennsylvania.**
Each node represents an ST which is labeled. Size of the nodes indicate sample size. Predicted founder groups are circled yellow. All connections are shown. The color of the edges indicates connection level, for instance black indicate SLVs, grey DLV etc. Connection levels between STs are also labeled; 1 for SLV, 2 for DLV etc.

to all antimicrobials evaluated, did not encode for toxin genes, and harbored two leucocidin genes, *lukAB* and *lukED*. The remainder of the seven isolates belonged to ST2187-2 through ST2187-8 variants. *S. aureus* ST2187-2 was resistant to vancomycin and did not encode for toxin or leucocidin genes. Variants ST2187-3 and ST2187-4 harbored *tsst-1* and *sec* and *sei* genes, respectively while ST2187-5 isolate was resistant to vancomycin and encoded *lukAB* and *lukED* genes. ST2187-6 harbored only *lukED* gene, while *S. aureus* isolate ST2187-7 was resistant to cefoxitin and the *S. aureus* isolate ST2178-8 was resistant to vancomycin and harbored only *lukAB* gene (Table 3).

A total of 28 isolates from 13 dairy herds were identified as ST3028, of which 14 and 14 isolates were from MM and BTM, respectively. *Staphylococcus aureus* isolates identified as ST3028 were categorized into 4 ST-variants based on antimicrobial resistance, toxin gene and leucocidin profiles. Three isolates of ST3028-1 were from cows with mastitis, the isolates were susceptible to all antimicrobials evaluated, did not encode for toxin genes, while harbored all three leucocidin genes, *lukMF', lukAB* and *lukED*. ST3028-2 was isolated from MM, the isolate encoded for *seg* and sei enterotoxin gens, and carried two leucocidin genes *lukAB* and *lukED*. ST3028-3 was the most predominant subtype (21 isolates, 75%) and was present in *S. aureus* isolates from 2008 (1 herd), 2013 (4 herds), 2014 (6 herds), and 2015 (5 herds). Isolates identified as ST3028-3 were susceptible to all antimicrobials evaluated, did not encode for toxin genes, and harbored two leucocidin genes, *lukAB* and *lukED* while *lukMF'* was not detected in this ST-variant. Three isolates of *S. aureus* ST3028-4 were isolated from BTM from two dairy herds, the isolates were resistant to vancomycin, and did not encode for toxin genes and harbored two leucocidin genes, *lukAB* and *lukED*. Nine isolates of ST698 from one farm were isolates from nine different cows with mastitis. All nine isolates were sensitive to all antimicrobials, did not encode for enterotoxins and lacked *lukMF'* gene. ST3273 and ST3274 were both isolated from BTM and are reported for the first time (Table 3).

A total of 37 isolates from 11 dairy herds were identified as ST151, of which 31 and six isolates were from MM and BTM, respectively. ST151 were categorized into 8 ST-variants based on antimicrobial resistance, toxin gene and leucocidin profiles. ST151-1 was the most predominant variant (28 isolates, 76%) and was isolated in 2008 (3 herds), 2013 (2 herds), 2014 (1 herd), and 2015 (6 herds). The 28 isolates identified as ST151-1 were susceptible to all antimicrobials tested, encoded for *seg* and *sei* enterotoxin genes and harbored all three leucocidins genes (*lukMF', lukAB* and *lukED*). The remainder of the nine isolates belonged to ST151-2 through ST151-8 variants. Five of the nine isolates were resistant to one or two antimicrobials, all the isolates with the exception of ST151-8, that encoded for *seg* and *sei*, while in addition two isolates, one each encoded for enterotoxin *sen*, and *sed*. Eight isolates encoded for all three leucocidins including *LukMF'*, while ST151-4 lacked *lukMF'*. Two isolates of ST705, one from 2008 and other from 2014 from two different farms were isolated from cows with mastitis. With the exception of the 2014 which was resistant to penicillin, both isolates had identical enterotoxin (*sec, seg, sei, tsst-1, sel)* profile and encoded for all three leucocidin genes. ST3272 was reported for the first time and was isolated from a cow with mastitis, encoded for *seg*, and *sei* enterotoxins and all three leucocidin genes (Table 4).

Six CCs comprising of ST1, ST15, ST45, ST72, ST350, and ST398 were infrequently isolated from cows with mastitis and BTM. One ST15 isolate from BTM encoded for enterotoxin *seh*, *lukAB* and *lukED*, while two isolates of ST15 were isolated from cow with mastitis and were resistant to penicillin, and as observed with ST1, lacked *lukMF'* gene. ST45, ST72 and two ST350, encoded for some of the genes of the *egc* cluster, all isolates; except for one isolate of ST350 lacked *lukMF'* gene. Two isolates of ST398 one isolated in 2014 and the other in 2015 were both resistant to four antimicrobials, while isolate encoded for *lukAB* only, the other lacked all three leucocidin genes (Table 5).

Six herds were selected to illustrate the prevalence and distribution of STs within a given herd (Table 6). A total of seven isolates collected in the month of August 2014 from BTM for herd 2, showed the presence of four STs (352–8, 3028–3, 3028–4, 3273) of which ST variant ST3028-3 accounted for four of the seven isolates. In herd 3, a total of six BTM samples collected in the month of September of 2014, belonged to two ST3028-3 and ST3028-4, the only difference between the two variants was resistance to vancomycin in ST3028-4. In herd 24, all four BTM isolates were from the month of August 2014, accounting for ST2187-1 and ST2187-7, as seen with herd 3, the only difference between the two variants was resistance to cefoxitin in ST2187-7. In herd 36, a total of four (151–1, 151–7, 2187–1, 352–7) ST variants were identified, of which ST151-1, was isolated from BTM, and three cows, of which one cow

**Table 6. Distribution of sequence types on six dairy herds in Pennsylvania.**

| Herd No. | Date | Source[a] (Cow No., Quarter) | Sequence Type Profile[b] | Anti-microbial resistance[c] | Toxin gene profile | Leucocidin profile |
|---|---|---|---|---|---|---|
| 2 | 8-4-2014 | BTM | 3028–3 | - | - | *LukAB2-lukED* |
| | 8-11-2014 | BTM | 3028–3 | - | - | *LukAB2-lukED* |
| | 8-18-2014 | BTM | 352–8 | VA | - | *LukMF'-LukAB2-lukED* |
| | 8-18-2014 | BTM | 3028–4 | VA | - | *LukAB2-lukED* |
| | 8-18-2014 | BTM | 3028–3 | - | - | *LukAB2-lukED* |
| | 8-18-2014 | BTM | 3273 | - | - | *LukMF'-LukAB2-lukED* |
| | 8-25-2014 | BTM | 3028–3 | - | - | *LukAB2-lukED* |
| 3 | 9-8-2014 | BTM | 3028–3 | - | - | *LukAB2-lukED* |
| | 9-8-2014 | BTM | 3028–4 | VA | - | *LukAB2-lukED* |
| | 9-15-2014 | BTM | 3028–3 | - | - | *LukAB2-lukED* |
| | 9-23-2014 | BTM | 3028–3 | - | - | *LukAB2-lukED* |
| | 9-23-2014 | BTM | 3028–3 | - | - | *LukAB2-lukED* |
| | 9-23-2014 | BTM | 3028–4 | VA | - | *LukAB2-lukED* |
| 24 | 8-4-2014 | BTM | 2187–1 | - | - | *LukAB2-lukED* |
| | 8-18-2014 | BTM | 2187–1 | - | - | *LukAB2-lukED* |
| | 8-25-2014 | BTM | 2187–7 | FOX | - | *LukAB2-lukED* |
| | 8-25-2014 | BTM | 2187–1 | - | - | *LukAB2-lukED* |
| 36 | 3-25-2015 | BTM | 151–1 | - | *seg, sei* | *LukMF'-LukAB2-lukED* |
| | 4-7-2015 | MM (451-LR) | 151–7 | DA, ERY | *seg, sei* | *LukMF'-LukAB2-lukED* |
| | 4-7-2015 | MM (890-RR) | 151–1 | - | *seg, sei* | *LukMF'-LukAB2-lukED* |
| | 4-7-2015 | MM (2389) | 151–1 | - | *seg, sei* | *LukMF'-LukAB2-lukED* |
| | 4-7-2015 | MM (567) | 151–1 | - | *seg, sei* | *LukMF'-LukAB2-lukED* |
| | 4-17-2015 | MM (890-LF) | 151–1 | - | *seg, sei* | *LukMF'-LukAB2-lukED* |
| | 5-15-2015 | MM (1024) | 2187–1 | - | - | *LukAB2-lukED* |
| | 7-17-2015 | MM (735) | 352–7 | TET | - | *LukMF'-LukAB2-lukED* |
| | 8-13-2015 | MM (451-LR) | 151–7 | DA, ERY | *seg, sei* | *LukMF'-LukAB2-lukED* |
| | 8-13-2015 | MM (3488) | 352–7 | TET | - | *LukMF'-LukAB2-lukED* |
| 51 | 10-11-2013 | MM (2094-RR) | 2187–1 | - | - | *LukAB2-lukED* |
| | 10-11-2013 | MM (2038-LR) | 3028–1 | - | - | *LukAB2-lukED* |
| | 10-21-2013 | MM (2028, RF) | 2187–1 | - | - | *lukAB2-lukED* |
| | 10-21-2013 | MM (2028, LF) | 3028–1 | - | - | *lukAB2-lukED* |
| | 10-21-2013 | MM (2038-LR) | 2187–1 | - | - | *lukAB2-lukED* |
| | 10-21-2013 | MM (2094-LF) | 2187–1 | - | - | *lukAB2-lukED* |
| | 10-21-2013 | MM (94) | 2187–1 | - | - | *lukAB2-lukED* |
| | 5-14-2015 | MM (2292) | 3028–1 | - | - | *lukAB2-lukED* |
| | 7-16-2015 | MM (2318) | 3028–2 | - | - | *LukMF'-LukAB2-lukED* |

*(Continued)*

**Table 6.** (Continued)

| Herd No. | Date | Sourceª (Cow No., Quarter) | Sequence Type Profileᵇ | Anti-microbial resistanceᶜ | Toxin gene profile | Leucocidin profile |
|---|---|---|---|---|---|---|
| 74 | 5-14-2008 | MM (7375) | 45 | PEN | *seg, sei, sem, seo, sen* | *lukAB2* |
| | 5-23-2008 | MM (1881) | 3028–3 | - | *seg, sei* | *lukAB2-lukED* |
| | 8-26-2013 | MM (4150) | 151–1 | - | *seg, sei* | *LukMF'-LukAB2-lukED* |
| | 7-9-2014 | MM (4159, RF) | 3028–3 | - | - | *lukAB2-lukED* |
| | 8-27-2014 | MM (4513) | 350 | - | *sei, sem* | *lukAB2* |
| | 8-27-2014 | MM (4159, RF) | 3028–3 | - | - | *lukAB2-lukED* |
| | 1-19-2015 | MM (4729) | 151–1 | - | *seg, sei* | *LukMF'-LukAB2-lukED* |
| | 4-1-2015 | MM (4120, LF) | 151–1 | - | *seg, sei* | *LukMF'-LukAB2-lukED* |
| | 4-1-2015 | MM (4120, LR) | 151–1 | - | *seg, sei* | *LukMF'-LukAB2-lukED* |
| | 4-1-2015 | MM (4316) | 151–1 | - | *seg, sei* | *LukMF'-LukAB2-lukED* |
| | 4-1-2015 | MM (4426) | 151–1 | - | *seg, sei* | *LukMF'-LukAB2-lukED* |
| | 4-1-2015 | MM (4617) | 151–1 | - | *seg, sei* | *LukMF'-LukAB2-lukED* |
| | 4-1-2015 | MM (4977) | 151–1 | - | *seg, sei* | *LukMF'-LukAB2-lukED* |
| | 4-1-2015 | MM (4986) | 151–1 | - | *seg, sei* | *LukMF'-LukAB2-lukED* |
| | 4-1-2015 | MM (7642) | 151–1 | - | *seg, sei* | *LukMF'-LukAB2-lukED* |
| | 4-1-2015 | MM (4589) | 151–1 | - | *seg, sei* | *LukMF'-LukAB2-lukED* |
| | 10-27-2015 | MM (4756) | 151–1 | - | *seg, sei* | *LukMF'-LukAB2-lukED* |
| | 10-27-2015 | MM (5163) | 2187–1 | - | - | *lukAB2-lukED* |

ª Sample: BTM: bulk tank milk; MM; Mastitic milk (Cow No, quarter: RF, right front; RR, right rear, LF, left front, LR, left rear).

ᵇ Sequence Type Profile: composite profile including ST, antimicrobial resistance, toxin genes and leucocidin genes (e.g., ST151-1; -, *seg, sei, lukMF'-lukAB-lukED)*.

ᶜ Antimicrobial: DA Clindamycin; ERY, Erythromycin; PEN, Penicillin; VA, Vancomycin.

(890) had two quarters infected with ST151-1 ten days apart. Similarly, ST variant 151–7 isolated in April of 2015 in left rear quarter of cow 451, and eight months later in August 2015 the same quarter was infected with ST151-7 (Table 6).

A review of nine isolates from herd 51 from 2013 (7 isolates) and 2015 (2 isolates) showed that ST2187-1 and ST3028-1 were isolated in 2013, while in 2015, ST3028-1 and ST3028-2 was identified. ST2187-1 was observed in five cows, of which cow 2094, had two different quarters infected over a 10-day period. Cow 2028 had two different quarters infected with two different ST variants, ST2187-1 and 3028–1 (Table 6).

Isolates from herd 74 were all from cows with mastitis. The 18 isolates were from 2008 (n = 2), 2013 (n = 1), 2004 (n = 3), and 2014 (n = 12). Over this period five STs including ST45, ST3028, ST350, ST151, and ST2187 were identified. The two isolates from 2008, were ST45 and ST3028-3. ST3028-3 was also observed in 2014 from one cow with two infected quarters, while ST350 was isolated from one cow in the same year. ST151-1 was first observed in 2013 and was later isolated in 2014 from 10 cows with mastitis. ST2187-1 was observed in 2015. (Table 6).

## Discussion

Molecular diagnostic tools such as PCR assays for virulence genes, pulsed field gel electrophoresis, multi-locus sequence typing and whole genome sequencing have made it not only possible to understand the pathogenomics of infectious agents, but also learn about their phylogenetic and evolutionary relationships. More so, MLST is now widely used to subtype and decipher the phylogenetic and evolutionary trends of several bacterial pathogens including *S. aureus*.

*Staphylococcus aureus* CC97 is of bovine origin and to-date has been reported from 18 countries (S2 Table). These reports show that ST97 and ST352, are more widely distributed in many countries and these two lineages of CC97 have well adapted to the bovine host [17, 20], while distribution of ST693, ST2187 and ST3028 appears to be restricted to United States and Canada. To-date two novel STs, ST3273 and ST3274 belonging to CC97 have not yet been reported from other countries. Based on these observations it can said that CC97 is distributed globally, and the STs associated with this clonal complex exhibit considerable diversity from country to country and within a region of a country (S2 Table).

The second largest CC observed in this study was CC151 and comprised of three STs, ST151, ST705, and ST3272. CC151 and its STs, have been reported from milk samples from 10 countries (S2 Table). The most prevalent ST in these ten countries was ST151. There are few studies that has identified ST151 or ST705 as belonging to CC705 [2, 15, 32]. A review of the database https://pubmlst.org/organisms, revealed that ST151 or ST705 have not been assigned to any specific clonal complex. The assignment of a ST to a CC, largely depends on the existing sequences in the database and the sequences and must meet the criteria of having at least four or more loci, with the exception that they more closely match another central genotype [33]. This could have resulted in identifying one ST as the founder and the other as its subgroup and this could explain the discrepancy associated with ST151 and ST705. In our study Grape-Tree analysis assigned ST3272 to CC151, however this ST has yet to be assigned to a CC in the *S. aureus* MLST database (https://pubmlst.org/organisms,). Application of Bayesian phylogenetic analyses indicated that CC97 and CC151 originated from human and adapted to bovines, to be now identified as bovine origin STs [34].

Assessment of the genetic variation at individual loci showed that locus *aroE* in comparison to the other six housekeeping gene fragments showed the highest genetic variation at an individual MLST locus. The *aroE* locus had many more allelic types (n = 10), with a Simpson's diversity index of 0.933 and had the highest number of segregating sites/polymorphic sites (n = 11), signifying non-synonymous substitution mutations. The diversity observed could be due to the location of the locus near an insertion site of a mobile element permitting frequent opportunities for recombination [17]. The diversity of CC97 can be attributed to recombination allowing clonal diversification [35]. The *yqiL* locus had 19 nucleotide polymorphic sites which translated to only 5 in amino acid sequence, identified as synonymous substitution. Overall based on nucleotide diversity among the seven loci and Tajima's D values, the nucleotide diversity was not statistically significant for any of the genes indicating that evolution could have occurred through random process. The non-synonymous to synonymous mutation ratio of <1, suggests that negative selection could have resulted in removing deleterious genetic polymorphisms through random mutations (Table 2).

The clonality of *S. aureus* isolates from Pennsylvania ranged from 0.4 to 0.7 suggestive of the fact that the CCs identified are clonal in structure. This observation concurs with that reported by Pérez-Montarelo et al. [36] showed that *S. aureus* isolates belong to a highly clonal population consisting of closely related CCs. However, the high degree of clonality does not explain the diversity in virulence factors, and the overall ability of the CCs to cause disease [36].

*Staphylococcus aureus* ST1, was isolated from a BTM, while ST15, ST45, and ST72 were isolated from cows with mastitis, and all of these STs have been identified as having human lineages. Feil et al. (2003) reported CC1, CC5, CC8, CC9, CC12, CC15, CC22, CC30, CC45 and CC51 were of human lineages. The STs including ST1, ST15, ST45 and ST72 have been reported from cows with mastitis and their caretakers in Algeria (CC1; [37]), Switzerland (CC1, CC15, CC45; [38, 39]), bovine milk or clinical samples in Australia (ST1, ST15; [2]), Brazil (ST1; [40]), China (ST1; [12, 41]), Denmark (ST1, ST15; [42]), Ethiopia (ST1; [43]),

Germany (ST45; [44]), Iran (ST15; [45]), India (ST72; [46]), Italy (CC15; [47]), Japan (ST1, ST72; [15]), Norway (CC1; [48]), South Africa (CC45; [13]), United Kingdom (ST1; [49]), United States (CC15, CC45, ST1; [32, 50]) (S2 Table).

Sakwinska et al. [39] inferred that human lineages of *S. aureus* are less likely to be found in animals and it's not unusual to find few human *S. aureus* STs present among bovine mastitis samples, as this could be due to contamination of bovine milk samples with *S. aureus* shed by humans while handling or processing milk. This explains the low frequency of occurrence of *S. aureus* human lineage STs, or the lack of widespread adaptation in bovines.

In our study, *S. aureus* ST350 was isolated from a cow with mastitis, this ST has also been reported by Smith et al. [20] from a cow with mastitis from a herd from New York. This ST has not been previously observed from other countries (Table 1). In Pennsylvania, ST398 was isolated from a cow with mastitis and from a BTM. This ST has been previously reported in China [41, 51] and milk samples from Germany [52].

The GrapeTree analysis of *S. aureus* STs, from Pennsylvania revealed that the majority of the STs were grouped into two CCs, CC97 and CC151. A similar GrapeTree distribution was reported with *S. aureus* isolates from BTM in Vermont [32]. GrapeTree analysis identified all the STs found in Pennsylvania and also that present in other regions (Washington State, New York, and Vermont) of the United States. The following STs CC15(ST25), CC30(ST30), CC8 (ST8), CC20(ST20), CC45(ST3023) CC97 (ST2189, ST3020, ST3021, ST3022, ST3024, ST3206, ST3027) and CC705 (ST2185) [20, 32, 50] have been reported in the United States, but however were not present in milk samples from Pennsylvania.

We also compared our data to the *S. aureus* STs in the world database for isolates from bovine milk. The GrapeTree analysis identified ten CCs, as primary founders. CC97 (subgroups ST352 and ST2187) is a major clonal population of *S. aureus* reported from bovine milk samples worldwide. CC151 is a quickly developing clonal complex of which ST705 is a subgroup founder. CC15 (ST15) was the only CC and ST that was found in Pennsylvania from a cow with mastitis and not reported in the world *S. aureus* MLST database associated with bovine milk, although CC15(ST15) have been recently reported from Iran [45]. CC1 from our data also represents an evolving bovine associated *S. aureus* group in the world. Other CCs like CC5, CC8, CC20, CC25, CC479, and CC504 from the world dataset were not represented in our isolates from Pennsylvania.

Nearly 80.3% of *S. aureus* isolates in this study were susceptible to all 10 antibiotics evaluated. A study conducted in Italy also found 80% (230 of 285) of the *S. aureus* isolates from milk were susceptible to all antibiotics evaluated [40], while a study reported from India found that 5% of the *S. aureus* isolates were pan-susceptible [46].

In our study, 8.5% of the isolates were resistant to beta-lactams (penicillin and cefoxitin); four isolates of *S. aureus* were of human lineage (ST15, ST45 and ST72) and six isolates of bovine lineage, ST97 and ST398. None of the isolates belonging to ST151 showed resistance to penicillin. Sakwinska et al. [39] observed that penicillin resistance in Swiss and French CC97 *Staph. aureus* isolates showed a higher frequency of resistance than those isolates belonging to CC151 and CC20.

In Hungary, Peles et al. [53] found that resistance to penicillin among *S. aureus* isolates from cows with mastitis was 88.9% and 20% from those from BTM. Rola et al. [54] found 38% of *S. aureus* from dairy operations in Poland were resistant to penicillin. A study conducted in Iran showed that 91.3% of S. aureus isolates from cows with mastitis were resistant to penicillin [45]. Li et al. [51] found that 90.4% of CC97 *S. aureus* isolates from cows with mastitis showed resistance to penicillin, while another study from northwestern China found that 80.5% of *S. aureus* from bovine milk were resistant to penicillin [55]. Jørgensen et al. [48] found that 7.7% of *S. aureus* isolates from bovine milk in Norway were resistant to penicillin. Based on these

reports it can be inferred resistance to penicillin among *S. aureus* isolates is a common occurrence on dairy herds and the frequency of penicillin resistance varies among countries.

In our study resistance to erythromycin and clindamycin was observed in 1.8% and 2.4% of isolates, respectively. Erythromycin resistant strains of *S. aureus* isolates accounted for 5.2% (China), 34.8% (Iran), 64.10% (India) 15.7% (China) of *S. aureus* isolates obtained from bovine mastitis or milk samples [41, 45, 46, 51]. While 6, 6, 19, 52, and 79.6%, clindamycin resistant strains of *S. aureus* isolates were reported from China [51], Austria [56], China [57], Brazil [58], and India [46], respectively. Resistance to erythromycin is determined by the *erm* gene and its variants or through mediated efflux pumps. The expression of the *erm* gene results in a product called as ribosome methylase however mutations in the promoter region of *erm* gene allows production of methylase that confers resistance to both erythromycin and clindamycin [59].

Resistance to tetracycline has been widely reported among *S. aureus* isolates from bovine mastitis and BTM. Resistance to tetracycline was observed with 3, 5.2, 18, 30.4, and 41%, strains of *S. aureus* isolates from Poland [54], European Union [60], China [57], Iran [45], and India [46], respectively. The findings our study (3.7% resistance to tetracycline) is comparable to that observed with European Union, which reflects on similar dairy operations and management in the US and the European Union and laws that mandate prudent use of antimicrobials.

Eleven isolates (6.74%) were resistant to vancomycin, and these isolates belonged to CC97 (ST352, ST2187, and ST3028) and CC151(ST151). Several studies conducted thus far have shown that *S. aureus* isolates of bovine origin are susceptible to vancomycin [2, 46, 61]. A recent study conducted in China found 3% of isolates showed resistance to vancomycin [57], while a study from Ethiopia reported 73.3% of *S. aureus* isolates were resistant to vancomycin [62]. Sung and Lindsay [63] tested a model for vancomycin resistance transfer from *Enterococcus faecalis* to *S. aureus* ST15 and showed that ST151 isolate was hypersusceptible to the acquisition of vancomycin-resistance genes from *Enterococcus sp*. This finding is of concern with respect to microbiomes in hospital and food production environments that may comprise of vancomycin resistant *Enterococcus* spp. and *S. aureus* STs such as 151, as this provides an opportunity for transfer of vancomycin resistant genes and result in emergence of vancomycin resistant *S. aureus* strains. Vancomycin resistant *S. aureus* STs, in ST151 in BTM and cows with mastitis pose a public health risk and needs to be further monitored and investigated.

Two isolates of ST398 showed multidrug resistance (>3 antibiotics) against clindamycin, erythromycin, tetracycline, and penicillin. Wang et al. [41] also observed ST398 isolates from cows with mastitis exhibited two MDR patterns of resistance (penicillin, ciprofloxacin, enrofloxacin, erythromycin and tilimicosin; penicillin, chloramphenicol, and gentamycin). ST398 has been found in pigs and horses, and MRSA variants of ST398 have been isolated from humans, cows, pigs and horses in Europe and China, which indicates that ST398 may have a broader host range [39, 64].

Based on the findings in this study and that reported from many other countries suggest that the frequency of occurrence antimicrobial resistant *S. aureus* isolates in a dairy operation varies between countries and region. This variation could be ascribed to the type and frequency of use of antimicrobials, the advancement and intensity of dairy practices, national regulations on approved use of antimicrobials, and education and awareness of producers and veterinarians on prudent use of antimicrobials [65].

There are >23 staphylococcal toxins listed by the International Nomenclature Committee which include, the toxic shock syndrome toxin (*tsst-1*), staphylococcal enterotoxins (*sea* to *see*, *seg* to *sej*, *sel* to *seq* and *ser* to *set*) and staphylococcal superantigen-like (SSL) toxins (seik to seiq, seiu to seix) [10, 66]. Staphylococcal toxin (tsst-1) and the enterotoxins have been widely reported in *S. aureus* from cows with mastitis and from BTM. However, the frequency of the

occurrence of the toxins alone or in combination with other toxins varies from herd to herd and different geographical regions of the world [24, 40, 67–69].

Jarraud et al. [26] described the enterotoxin gene cluster, *egc* (*seg–sei–selm–seln–selo*), this cluster is located on the genomic island type II vSaβ. The enterotoxins expressed by the *egc* cluster are the most frequently reported enterotoxins from isolates of bovine origin. Genes of enterotoxin *egc*-cluster (*seg*, *sei*, *sem*, *seo*, and *sen)* were the most frequently identified enterotoxin genes in this study, of which, *seg/sei* was present in ~84.5% of enterotoxin gene positive isolates which is much higher that previously reported studies [69, 70]. Omoe et al. [24] investigated the distribution of enterotoxins from several sources including cows with mastitis and raw milk. Enterotoxins *seg* and *sei* were detected in 33% and 38.9% of isolates from cows with mastitis and raw milk, respectively. The study also identified 38.1% of the isolates from cows with mastitis were positive for *sec-seg- sei* profile. A similar profile was identified in this study.

The presence of *sec*, *tsst-1* or *sed* and *sej* gene combinations have been reported previously in *S. aureus* isolated from cows with mastitis and BTM samples [17, 67–69]. The results of this study showed that besides the traditional enterotoxin genes, *sed* and *tsst-1*, and the newly described genes seg, *seh*, *sei*, *sej*, *sel*, *sem*, *sen*, *seo*, *seq* and *ser*, were also detected in *S. aureus* isolate from MM and BTM. Haveri et al. [18] (2007) based on their study, suggested that the presence of *sed* and *sej* in an *S. aureus* isolated could be associated with persistent bovine mastitis. Matsunaga et al. [71] showed that *S. aureus* isolates that encoded for *sec* and *tsst-1* caused severe mastitis that were unresponsive to therapy. On the contrary, Tollersrud et al. [72] were unable to find a relationship between enterotoxin production different classifications of mastitis. Several studies have identified and characterized the role of mobile genetic elements such as pathogenicity islands, prophages and plasmids that encode for enterotoxins, antimicrobial resistant determinants and other virulence factors [24, 26, 73]. The diversity observed with respect to virulence factors and antimicrobial resistance in *S. aureus* strains can be explained by acquisition or loss of mobile genetic elements through horizontal transfer processes. To-date the role of staphylococcal toxins in the pathogenesis of bovine mastitis have not fully understood. It is suggested that enterotoxins enable the immunosuppression process thereby creating opportunities for *S. aureus* to persist and cause chronic infection. It's also likely that highly pathogenic strains are more likely to produce higher levels of toxins, as compared to those with low pathogenicity, and mere presence or absence of enterotoxin genes does not correlate with pathogenicity [71].

To-date, six leucocidins of *S. aureus* have been described which include γ -hemolysins (HlgAB and HlgCB) and LukAB located on the core genome, while LukED is present in most strains and is encoded on the pathogencity island ν Saβ [74]. The other two leucocidins are Panton Valentine Leukocidin and *lukMF'*, which are encoded on phages [44, 75].

Of the six leucocidins, *lukAB* is the most observed leucocidin in *S. aureus* isolates. LukAB is located on the core genome and not on mobile genetic elements. This could explain the high prevalence (96.3%) of *lukAB* in our study. Unlike other leucocidins, *lukAB* functionality is associated with integrin CD11b [76]. Vrieling et al. [25] showed that *lukAB* lysed human neutrophils while no effect was observed on bovine neutrophils. Based on these reports it can be inferred that *lukAB* may have limited role in the pathogenesis of *S. aureus* associated bovine mastitis.

The prevalence of *lukED* observed in this study (95%) is similar to that reported from Finland (96%), Japan (96%), and Spain (96.6%) and [18, 75, 77]. All leucocidins, with the exception of *lukAB*, bind to chemokine receptors. CXCR2 is the bovine receptor for *lukED*, but this leucocidin also interacts with CCR5 [78, 79]. *LukED* has the ability to permeabilize and kill cells such as neutrophils carrying the corresponding receptors [80]. Vrieling et al. [25] observed that supernatant of mastitis reference strain of *S. aureus*, Newbould 305, lacked

*lukMF'* operon and was unable to permeabilize the CCR1 expressing cells. However, the supernatant of the Newbould strain killed CXCR2 and C5aR1 expressing cells through secretion of functional HlgAB/LukED and HlgCB. The study also showed that invitro expression of *lukED* was highly variable among bovine mastitis isolates.

The prevalence of *lukMF'* observed in this study (54%) is similar to that reported from Germany (53.1 to 80%), Japan (62.6 to 86%), and Spain (50%) [44, 75, 77, 81]. Of the six leucocidins, isolates of *S. aureus* from cows with mastitis encode for the bi-component leucocidin, *lukMF'* [44, 75, 81]. Vrieling et al. [25] in their comprehensive study showed that *lukMF'* secretion and production by bovine mastitis isolates was strain related and was the primary toxin that was produced in enormous quantities that caused severe lysis of bovine neutrophils and monocytes. Their in-vitro and in-vivo studies showed that *lukMF'* expression levels correlated with neutrophil toxicity and was associated with severity of mastitis following experimental infection.

Combining STs, with toxin and leucocidin gene profiles and antimicrobial resistance profiles gives a better understanding of the molecular epidemiology of *S. aureus* at a cow level and herd level (within a herd) and between herds in Pennsylvania. For example, several ST-variants were present in BTM in milk samples collected at 7-14-day intervals from the same herd. ST-variants allowed tracking of repeat/ persistent infections in cows with infections in the same quarter or different quarters sampled over a period ranging from 10 days to 2 years.

A careful review of the distribution of ST-variants suggests that some *S. aureus* strains may have developed a preference to specific environments (e.g., mammary gland, milking system and bulk tank) in the dairy herd. For instance, *S. aureus* STs, such as ST352, ST3028 and ST2187 could have adapted to the cold environment of the bulk tank and milking system; this could have resulted in the emergence of ST-variants allowing the variants to colonize the bulk tank. Interestingly most of these isolates lacked enterotoxin and *LukMF'* genes. The mammary gland as an environment is preferred by most STs, and in our study, ST-variants belonging to CC97 and CC151 were isolated from cows with mastitis from several herds in 2008, 2013–2015. Based on the observations, it can be inferred that multiple STs can concurrently be prevalent in a herd over time. This is perhaps the first time in literature we are able to document that multiple ST-variants can be found in BTM and MM.

## Conclusions

The findings of our study show that small number of *S. aureus* STs types (ST352, ST2187, ST3028, and ST151) are associated with majority of cases of bovine mastitis in Pennsylvania. In addition, our study also observed in a herd, there could be one predominating ST of *S. aureus* which can coexist with several other ST types of *S. aureus* strains. However, when STs are interpreted along with virulence and leucocidin genes and antimicrobial resistance, ST-variants allowed better interpretation of the *S. aureus* molecular epidemiologic findings specifically tracing recurrence or persistence of infections in cow over time, among cows in the herd, and between herds in Pennsylvania.

## Supporting information

**S1 Table. List of primers and control strains used in the study.**
(DOCX)

**S2 Table. Clonal complexes and sequences types of *S. aureus* reported in literature [82–88].**
(DOCX)

## Acknowledgments

The authors acknowledge the laboratory and technical support provided by Katelyn Molinaro, Dr. Tatiana Laremore, Dr. Lingling Li, and Dr. Rashmi Satyakumar.

## Author Contributions

**Conceptualization:** Bhushan Jayarao.

**Data curation:** Asha Thomas, Shubhada Chothe, Subhashinie Kariyawasam, Erin Luley, Suresh Kuchipudi, Bhushan Jayarao.

**Formal analysis:** Asha Thomas, Shubhada Chothe, Maurice Byukusenge, Erin Luley, Suresh Kuchipudi.

**Funding acquisition:** Bhushan Jayarao.

**Investigation:** Asha Thomas, Shubhada Chothe, Maurice Byukusenge, Tammy Mathews, Traci Pierre, Erin Luley, Bhushan Jayarao.

**Methodology:** Traci Pierre.

**Supervision:** Subhashinie Kariyawasam, Bhushan Jayarao.

**Validation:** Erin Luley.

**Writing – original draft:** Asha Thomas, Shubhada Chothe.

**Writing – review & editing:** Maurice Byukusenge, Traci Pierre, Bhushan Jayarao.

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
