## [Decision Letter · Decision Letter 0]

12 Feb 2021

PONE-D-21-02556

Prevalence and distribution of multilocus sequence types of Staphylococcus aureus isolated from bulk tank milk and cows with mastitis in Pennsylvania.

PLOS ONE

Dear Dr. Jayarao,

Thank you for submitting your manuscript to PLOS ONE. After careful consideration, we feel that it has merit but does not fully meet PLOS ONE’s publication criteria as it currently stands. Therefore, we invite you to submit a revised version of the manuscript that addresses the points raised during the review process.

Your manuscript has been reviewed by an expert in your field.  A minor revision is suggested.

Please submit your revised manuscript by  2 weeks. If you will need more time than this to complete your revisions, please reply to this message or contact the journal office at plosone@plos.org. Please include the following items when submitting your revised manuscript:

We look forward to receiving your revised manuscript.

Kind regards,

Yung-Fu Chang

Academic Editor

PLOS ONE

Journal Requirements:

Reviewers' comments:

Reviewer's Responses to Questions

**Comments to the Author**

1. Is the manuscript technically sound, and do the data support the conclusions?

Reviewer #1: Yes

2. Has the statistical analysis been performed appropriately and rigorously? 

Reviewer #1: Yes

3. Have the authors made all data underlying the findings in their manuscript fully available?

Reviewer #1: Yes

4. Is the manuscript presented in an intelligible fashion and written in standard English?

Reviewer #1: Yes

5. Review Comments to the Author

Reviewer #1: In my opinion, the study entitled “Prevalence and distribution of multilocus sequence types of Staphylococcus aureus isolated from bulk tank milk and cows with mastitis in Pennsylvania” is interesting and valid for potential publication. It was conducted with appropriate methods and well written. A deep characterization of S. aureus strains isolated from cow’s milk was conducted, including MLST, antibiotic resistance and virulence factor.

Few minor concerns:

Bacterial isolates (LL123-130). It is not clear the criteria of strain selection. Authors should specify what were the inclusion criteria of the strains in the study. How did they avoid overrepresentation of potential clones?

This comment pairs with Clonality of S. aureus (LL 318-324). The results substantiate the clonal nature of S. aureus. Could this be related with the selection of the strains? Please clarify.

LL-4785-10: authors selected 6 herds out of 77 to illustrate the prevalence and distribution of ST within a given herd. What was the rationale of such decision? How they select the 6 herd? What is the possible inference of the information linked to the whole study?

Discussion section.

This section is very long and with very detailed information.

I suggest, maybe in the conclusion section to convey the key points of their findings:

- Significance of relationship between antibiotic resistance and/or virulence factor with origin of the strains (e.g. herd, type of milk);

- Significance of strains recurring over time

- Impact of such differences on a practical basis: treatment, mastitis control, possible transmission to humans, and so forth…

6. PLOS authors have the option to publish the peer review history of their article (what does this mean?). If published, this will include your full peer review and any attached files.

Reviewer #1: No

---

## [Author Response · Author response to Decision Letter 0]

23 Feb 2021

RESPONSE TO REVIEWER #1

The authors of this manuscript would like to take this opportunity to thank the reviewer for the in-depth review. The review provided has helped to improve the manuscript, more so with revising the conclusions of the study. Response to specific queries are as follows:

QUERY 1. 

Bacterial isolates (LL123-130). It is not clear the criteria of strain selection. Authors should specify what were the inclusion criteria of the strains in the study. How did they avoid overrepresentation of potential clones?

This comment pairs with Clonality of S. aureus (LL 318-324). The results substantiate the clonal nature of S. aureus. Could this be related with the selection of the strains? Please clarify.

RESPONSE:

The reviewer’s query is well founded. The investigators have revised the statements to clarify the inclusion criteria and avoidance of overrepresentation of potential clones.

Lines 128-136, have been revised and reads as follows:

Milk samples received for mastitis culture and bulk tank milk analysis in 2008 (n=500), 2013 (n=408), 2014 (n=265), and 2015 (n=279) were examined for S. aureus and other mastitis pathogens. A total of 181 S. aureus isolates were isolated and placed in the laboratory’s culture repository. Of the 181 isolates (11 isolates could not be recovered from the repository, and seven isolates had incomplete submission forms), 163 isolates from cows with mastitis (n=113 isolates) and from BTM (n=50) from 77 dairy herds in Pennsylvania were examined for their phenotypic and genotypic characteristics. As the focus of the study was to estimate the prevalence and distribution of S. aureus clonal types, control for over- or under-representation of S. aureus clonal types was not attempted.

Lines 128-136, have been revised and reads as follows:

The ISA for the S. aureus examined in this study (163 isolates comprising of 16 STs) was determined to be 0.7339 and 0.4201 indicating significant (p=0.000) linkage disequilibrium; suggestive of the clonal nature of S. aureus. The scale of clonality observed in this study could be attributed to the collection of S. aureus isolates analyzed in this study.

QUERY 2.

LL-4785-10: authors selected 6 herds out of 77 to illustrate the prevalence and distribution of ST within a given herd. What was the rationale of such decision? How they select the 6 herd? What is the possible inference of the information linked to the whole study?

RESPONSE:

The investigators included six herds, solely to illustrate a detailed picture of the distribution dynamics of ST over time. These six farms show: 1) re-infection of quarters in the cow with same ST over time, 2) multiple STs in the cow, 3) persistence of one ST in the herd over a period of 8 years, 4) ST-variants, and 5) occurrence of the same ST in infected quarters and bulk tank milk in a herd. 

Lines 481-482, have been revised and reads as follows

Six of the 77 herds were selected primarily to illustrate an in-depth picture of the distribution dynamics of STs overtime (Table 6).

QUERY 3. 

Discussion section.

This section is very long and with very detailed information. 

I suggest, maybe in the conclusion section to convey the key points of their findings:

- Significance of relationship between antibiotic resistance and/or virulence factor with origin of the strains (e.g., herd, type of milk);

- Significance of strains recurring over time

- Impact of such differences on a practical basis: treatment, mastitis control, possible transmission to humans, and so forth…

RESPONSE:

The authors would appreciate if the discussion section were retained in its current form. The reviewer’s suggestion to re-address the conclusion portion has in fact strengthened this paragraph. 

Lines 749-769, have been revised and reads as follows

The findings of our study show that small number of S. aureus STs types (ST352, ST2187, ST3028, and ST151) were associated with majority of cases of bovine mastitis in Pennsylvania. A substantial number of isolates belonging to CC151, encoded for both toxin genes ( i.e., seg, sei) and LukMF ’ leucocidin genes as compared to isolates from CC97. Antimicrobial resistance was not associated with any specific CC or ST, resistance to vancomycin was seen more frequently with S. aureus isolates from BTM. A careful review of the ST-variants showed that in a given herd, there could be one predominant ST of S. aureus, and this ST coexists with several other transient STs of S. aureus. STs when interpreted along with virulence and leucocidin genes and antimicrobial resistance, the ST-variants allowed better interpretation of the S. aureus molecular epidemiologic findings for tracing recurrence or persistence of infections in cow over time, among cows in the herd, and between herds in Pennsylvania. 

 For these findings to have practical applications for addressing herd and individual cow udder health (treatment, purchase of new stock, culling, mastitis prevention and control practices, including selection of strains for autogenous vaccines), herd managers and veterinarians should be encouraged to create a profile of S. aureus isolates prevalent in the herd, this information will assist in developing an evidence-based approach to prevent and control S. aureus bovine mastitis. Based on our experience doing this survey study, it is felt that veterinary diagnostic laboratories should develop and offer pathogen specific molecular epidemiology typing panels. These molecular epidemiologic typing panels, when coupled with traditional diagnostic tests, could result in the next generation of effective prevention and control practices for bovine mastitis.

---

## [Decision Letter · Decision Letter 1]

1 Mar 2021

Prevalence and distribution of multilocus sequence types of Staphylococcus aureus isolated from bulk tank milk and cows with mastitis in Pennsylvania.

PONE-D-21-02556R1

Dear Dr. Jayarao,

We’re pleased to inform you that your manuscript has been judged scientifically suitable for publication and will be formally accepted for publication once it meets all outstanding technical requirements.

Kind regards,

Yung-Fu Chang

Academic Editor

PLOS ONE

Additional Editor Comments (optional):

Reviewers' comments:

Reviewer's Responses to Questions

**Comments to the Author**

1. If the authors have adequately addressed your comments raised in a previous round of review and you feel that this manuscript is now acceptable for publication, you may indicate that here to bypass the “Comments to the Author” section, enter your conflict of interest statement in the “Confidential to Editor” section, and submit your "Accept" recommendation.

Reviewer #1: All comments have been addressed

2. Is the manuscript technically sound, and do the data support the conclusions?

Reviewer #1: Yes

3. Has the statistical analysis been performed appropriately and rigorously? 

Reviewer #1: Yes

4. Have the authors made all data underlying the findings in their manuscript fully available?

Reviewer #1: Yes

5. Is the manuscript presented in an intelligible fashion and written in standard English?

Reviewer #1: (No Response)

6. Review Comments to the Author

Reviewer #1: (No Response)

7. PLOS authors have the option to publish the peer review history of their article (what does this mean?). If published, this will include your full peer review and any attached files.

Reviewer #1: No

---

## [Editor Report · Acceptance letter]

3 Mar 2021

PONE-D-21-02556R1 

 Prevalence and distribution of multilocus sequence types of *Staphylococcus aureus* isolated from bulk tank milk and cows with mastitis in Pennsylvania. 

Dear Dr. Jayarao:

I'm pleased to inform you that your manuscript has been deemed suitable for publication in PLOS ONE. Congratulations! Your manuscript is now with our production department. 

Kind regards, 

on behalf of

Dr. Yung-Fu Chang 

Academic Editor

PLOS ONE